# Factors Influencing Public Trust in Open Government Data

Abdullah Almuqrin [1,*]🆔, Ibrahim Mutambik [1]🆔, Abdulaziz Alomran [1], Jeffrey Gauthier [2]
and Majed Abusharhah [3]

1   Department of Information Science, College of Arts, King Saud University, Riyadh 11451, Saudi Arabia
2   School of Business and Economics, State University of New York at Plattsburgh, Plattsburgh, NY 12901, USA
3   Ministry of Education, Jazan 82614, Saudi Arabia
*   Correspondence: aalmogren@ksu.edu.sa

**Abstract:** Open government data (OGD) involves exposing government data to the public, guided by the values of clarity, accountability, honesty, and integrity. This study investigates the impact of the perceived quality of data, systems, and services on citizens' trust in OGD, with the information systems success model as the theoretical framework. A questionnaire was delivered electronically to reach OGD users around the world. A total of 358 complete responses were obtained, representing 63.58% of all responses. Structural equation modeling was used to analyze the hypothesized relationships between constructs based on users' responses. The findings confirm the impact of data, system, and service quality on citizens' perceived trust in OGD. Moreover, perceived system and service quality had a positive impact on perceived data quality, and perceived service quality had a positive effect on perceived system quality. These findings indicate that OGD service quality affects data and system quality, making it the most fundamental motivator of citizens' trust in OGD. This highlights the role of open government platforms in developing public services and providing users with complete and correct data, feedback tools, and data visualization.

**Keywords:** open government data; trust; data quality; system quality; service quality; information systems success model; structural equation modeling

## 1. Introduction

The Internet and information technology have made it easier for citizens to access open government data (OGD). Such data have been made accessible to the public in accordance with the values of clarity, accountability, honesty, and integrity [1]. These values have influenced the relationship between public bodies and citizens, thus fostering a sense of trust [1]. Due to its recent inception, OGD has not been extensively debated and requires more research.

Most prior research has highlighted how citizens attain access to OGD and their trust in it. For instance, an extended technology acceptance model and the three principles of open government were used to survey the factors encouraging citizens in Germany to use OGD [2]. The positive factors detected were the principles of transparency, the ability to collaborate with government agencies, the ability to participate in government decision-making, ease of use, and the usefulness of open government. A study in Austria examined the use of online platforms for gaining citizens' trust to share in public decision-making. Citizens' use of an active online data platform increased their satisfaction and trust in their government and the data [3].

Trust has been associated with an open government but could depend on a citizen's perceived opportunity to share in decision-making and influence policy [4,5]. One study investigated whether citizens' trust could be influenced by OGD released by European governments. It used data from the 2018 Open Data Inventory and tested the impact of OGD on citizens' trust based on structural equation modeling. The more governments

expanded OGD and increased openness, coverage, and breakdown, the higher citizens' trust became [6].

Prior studies have assumed that citizens' trust in OGD can be won by high-quality OGD, platforms, and services, but not enough studies have empirically tested this postulation. Due to the importance of public trust in OGD, stakeholders need to understand the factors that can maximize this trust. Studies about the impact of OGD on citizens' trust have examined the government systems that deliver open data, ease of use and accessibility, and offer the possibility of giving feedback (e.g., [2,4]); government websites that provide information (e.g., [4]); or the features of the data themselves (e.g., [6]). Moreover, most studies have looked at the impact of people's trust in a government system on their OGD use rather than the impact of OGD use on their trust in this system, which is usually assumed more than empirically tested [7]. Delone and McLean's information systems success model was used to shape the theoretical framework of this study. This study contributes to the current understanding of the effectiveness of this model in addressing citizens' trust in OGD. This was achieved through investigating the relationships between the three quality attributes of data quality, system quality, and service quality and their impact on citizens' trust in OGD. Data quality is defined by [8] as "a reflection of the data accuracy, data truthfulness, data completeness, and data up-to-dateness of the sensed data" (p. 64267). System quality "measures system reliability and accessibility in regard to its performance of the required tasks" ([9], p. 2). Furthermore, service quality was defined by [10] as "A global judgement or attitude relating to a particular service; the customer's overall impression of the relative inferiority or superiority of the organization and its services. Service quality is a cognitive judgement" (p. 4).

Thus, this study has sought to answer the following questions:

1. Why are citizens willing to trust OGD?
2. How can the information systems success model facilitate citizens' trust in OGD?

## 2. Theoretical Framework

The theoretical framework was based on DeLone and McLean's information systems success model, a framework for measuring the success of information systems according to system quality, information quality, information use, user satisfaction, individual impact, and organizational impact [11]. The researchers updated this model to include net benefits instead of individual and organizational impact, in addition to service quality [12]. However, most researchers do not combine all the categories of this model in one study, using only one or two of them [7].

This study is one of the few to apply this model to open government and OGD by integrating the three attributes of data quality, system quality, and service quality. This section discusses the dependent variable (trust) and independent variables (data, system, and service quality). The relevant relationships between them are illustrated in Figure 1.

### 2.1. Citizens' Trust

Building public trust is one of the expected outcomes of effective OGD, which is an indication of good governance and constructive implementation of policies [4,13]. As a result, a high level of public trust is often seen as an indicator of the effectiveness of government services, while a lack of trust in OGD leads to the erosion of social cohesion [13]. Trust refers to "a psychological state that allows a person to accept vulnerability based upon positive expectations of the intentions or behavior of others" ([14], p. 1462).

There are three main mechanisms for trust-building: institutional-based trust, which refers to the commitment and confidence in formal societal systems; process-based trust, which involves a previous or expected exchange acquired by reputation or real experiences; and characteristic-based trust, which reflects personal characteristics [14]. When investigating trust in OGD, process-based trust is the most applicable since past experiences with the government system and the open data it delivers create trust. Accordingly, citizens' trust in OGD combines trust in the data themselves and the government that provides it.

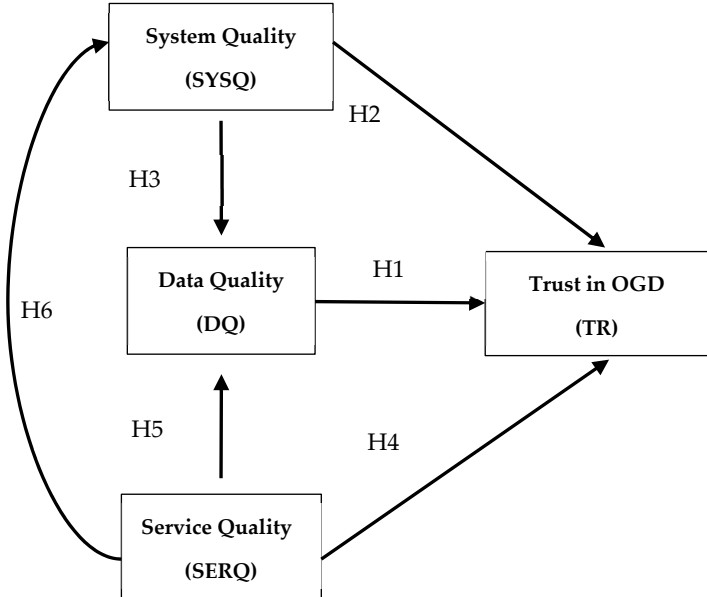

**Figure 1.** Theoretical framework.

### 2.2. Data Quality

While data quality is concerned with the technical features of data, information quality focuses on non-technical aspects [7]. More specifically, data quality indicates how complete, correct, accurate, and up-to-date data are considered to be, while information quality is information's "accuracy, details, timeliness, and validity" ([8], p. 64267). In this study, data quality is used to describe both technical and non-technical features of data. Some users look for accurate data, while others are more interested in relevant, up-to-date data. This spectrum highlights the importance of data quality in open government and its significance in OGD implementation.

Most studies have found poor data quality to be a challenge facing governments, with a negative effect on trust in data and the government providing it (e.g., [15–17]). The authors of [16] argued that accuracy, completeness, timeliness, understandability, and consistency are the main aspects of high data quality, thus contributing to higher trust. This led to the first hypothesis of the present study:

**H1.** *Perceived data quality will have a significant impact on citizens' trust in OGD.*

### 2.3. System Quality

System quality is one of the core elements of DeLone and McLean's information systems success model, which measures a system's reliability in terms of accessibility, ease of use, and response time [9,18]. System quality is closely related to users' trust in a system since reliable technical aspects of a system affect trust in the system's constituents and the services provided [17,19]. Open government systems provide portals to engage citizens in government datasets by enabling them to search and download materials, among other practices. The authors of [20] found the response time to be the foundation for assessing a system's performance. Moreover, the technique used by the system to publish the OGD determines citizens' approach to using that data [2]. Consequently, higher system quality—represented in its availability, performance, responsiveness, and documentation—can increase citizens' trust in OGD, leading to the second hypothesis of this study:

**H2.** *Perceived quality of an open government system will have a significant impact on citizens' trust in OGD.*

OGD portals are not only an essential part of open data infrastructure but are also a platform for data management that includes tools facilitating relevant data search and

manipulation [21]. Data portals contain metadata as part of the documentation that provides information about the data structure and acts as an incentive for citizens to use and understand these datasets. In addition, the functionalities and performance of open data portals can improve citizens' perception of data quality by facilitating data representation, differentiation, and association. This consideration led to the third hypothesis:

**H3.** *Perceived open government system quality will have a significant impact on perceived data quality.*

*2.4. Service Quality*

From a user's point of view, OGD service quality is concerned with providing exclusive public services that meet the expectations of citizens when using open government portals and websites. This can be achieved by providing effective data streams and enabling communication between citizens and governments [22]. People are more satisfied when services meet their expectations [23], and the higher the quality of services that provide value, the higher growth of citizens' trust, facilitating continued OGD use [22].

A large number of service quality models were introduced. For example, the SERVQUAL model was introduced by [24] to measure customer perception of service quality based on five major dimensions. These dimensions include assurance, empathy, responsiveness, reliability, and tangibles. All five dimensions have been documented to influence citizens' perceptions of service quality [22,25,26]. The items of the SERVQUAL scale were subject to some modifications, such as negatively rewording six items, positively rewording 16 items, and substituting some items with other items [24].

Another service quality model, the IT alignment model, was proposed by [27]. This model emphasizes the role of information technology in improving specific dimensions of service quality such as communications, security, competence, reliability, access, responsiveness, and knowing the customers. The concept of this model is based on closely aligning and coordinating strategies of information systems with service quality [27]. Moreover, the service quality, customer value, and customer satisfaction models was proposed by [28]. This integrative model incorporates customer satisfaction, perceptions, customer value, intention to repurchase, and service quality. The model can be beneficial in understanding and interpreting a customer's decision-making with an emphasis on customer value and service quality [28]. Good and satisfying service quality is expected to raise public trust in OGD, leading to the fourth hypothesis:

**H4.** *Perceived open government service quality will have a significant impact on citizens' trust in OGD.*

Studies have suggested that factors such as information, data, and system quality have an indirect influence on service quality. For example, [23] suggested integrating information and system quality with service quality to ensure that service quality expectations are aligned between users and service providers. The authors of [22] divided service quality into two main parts: technical performance quality, which is affected by system quality among other factors supporting service delivery, and service function quality, which is influenced by information quality in addition to other factors that facilitate online services. However, this study argues that higher service quality can improve OGD quality. Furthermore, it is suggested that when OGD services are assuring, empathic, responsive, and reliable, users will perceive higher data and system quality. In particular, service providers who are emphatic and responsive will seek to enhance data quality by responding to user inquiries and fixing inaccurate data. In addition, service providers can enhance the user experience with OGD by ensuring easy access to data. These assumptions led to the fifth and sixth hypotheses:

**H5.** *Perceived open government service quality will have a significant impact on perceived OGD quality.*

**H6.** *Perceived open government service quality will have a significant impact on perceived open government system quality.*

## 3. Materials and Methods

According to DeLone and McLean's information systems success model, many factors affect the ability of citizens to trust open government and OGD [11]. Under that framework, this study employed an online questionnaire to assess the impact of data, system, and service quality on citizens' trust in OGD; the impact of service quality on data and system quality; and the impact of system quality on data quality. The questionnaire is in clear English language, and it will take users from 7–10 min to complete.

### 3.1. Instrument

The questionnaire, which surveyed respondents' opinions and demographic information, was developed from validated scales in the literature, as shown in Table 1. Slight modifications were made to fit the context of the study. All items used were examined for readability, suitability, consistency, and face validity with the help of a panel of five faculty members specialized in information technology and data science. Moreover, a pilot study was conducted with 23 employees in a Saudi open data office. Their responses were taken into consideration to increase readability and clarity. Responses were measured on a 5-point Likert scale ranging from "strongly disagree" to "strongly agree," in addition to a "not applicable" choice. Although the Likert scale is one of the most universal and trusted methods that is widely used in measuring attitude, attitude might be affected by the space between the five choice options, which is not equidistant. In addition, respondents' answers might be influenced by prior questions answered. Nevertheless, a 5-point Likert scale was used to increase response rate and quality while reducing respondents' "frustration level" [29,30]. This scale makes it simple for participants to read out the complete list of scale descriptors (1: strongly agree, etc.) [31].

**Table 1.** Constructs.

| Construct | Definition | Items | Source |
|---|---|---|---|
| Data Quality (DQ) | A reflection of the data's perceived accuracy, truthfulness, completeness, and up-to-dateness | 2 | ([8], p. 64267) |
| System Quality (SYSQ) | The reliability of the system in terms of online response time, ease of use, and accuracy | 4 | ([18], p. 313) |
| Service Quality (SERQ) | How well online public services provided by government websites meet the user's requirements and the extent to which government websites facilitate efficient and effective information search and online transactions as well as communication between government and citizens | 4 | ([22], p. 2) |
| Trust (TR) | A psychological state that allows a person to accept vulnerability based on positive expectations of the intentions or behavior of others | 3 | ([14], p. 1462) |

### 3.2. Data Collection

The questionnaire was created using Google Forms, and its link was distributed from January to February 2022 through multiple open data community platforms, including Facebook, WhatsApp, LinkedIn, Twitter, and available mailing lists. The questionnaire was available to anyone in the world. An electronic consent form was signed by each respondent before starting the questionnaire. Convenience sampling and snowball sampling were used for recruitment to avoid barriers that would make reaching potential OGD users more challenging.

Data collection yielded 563 responses; of these, 147 included unanswered questions, 10 answered "not applicable" for questions about trust, and 48 answered "not applicable" for questions about data, system, and service quality. The invalid responses were removed from the sample, as recommended by [32]. Accordingly, 205 responses were excluded,

and only 358 complete responses were used in the data analysis, representing 63.58% of all responses.

### 3.3. Respondent Demographics

The demographics of the accepted respondents are presented in Table 2.

**Table 2.** Respondent demographics.

| Factor | Sub-Factor | Percent |
|---|---|---|
| Age | Average = 33 years | |
| Gender | Female | 50.9% |
| | Male | 49.1% |
| | Art | 10.4% |
| | Computer Science | 14.9% |
| | Education | 14.3% |
| | Engineering | 10.7% |
| Major | Management | 12% |
| | Medical | 10.5% |
| | Science | 22.7% |
| | Other | 4.2% |
| | less than 1 | 0.4% |
| Years of Experience with OGD | 1 to less than 3 | 5.5% |
| | 3 to less than 5 | 20.7% |
| | 5 or more | 73.4% |
| | Middle East | 15.3% |
| | Africa | 18.6% |
| Country/Region of Origin | U.S. | 25.7% |
| | Europe | 20.1% |
| | Asia | 10.4% |
| | Other | 9.9% |

Slightly over half the respondents were women, and the average age was 33. In terms of experience, 73.4% had engaged with OGD for five years or more, suggesting a large number of respondents had experience using OGD and open government platforms. Moreover, 209 (58.38%) had used OGD as part of a team. The highest-ranked use of the data was creating applications and visualizations, including graphics and maps. The U.S. was the place of origin with the most respondents (25.7%), followed by Europe (20.1%). However, the findings revealed no significant differences in nationality concerning trust in OGD or perceptions of data, system, and service quality.

### 3.4. Data Analysis

The data collected were analyzed following Anderson and Gerbing's recommendation to use a two-step approach: one based on the measurement model and the other testing the structural relationships between latent variables [33]. This made it possible to evaluate the validity and reliability of constructs before using them in the main study. Next, the estimates of the structural model were investigated using the coefficients of determination ($R^2$ values) and the significance of the path coefficients. By using structural equation modeling, the study assessed hypothesized influences of independent variables on dependent variables, i.e., the structural model, and evaluated the loadings of several constructs (indicators) on their predicted latent constructs, i.e., the measurement model [32]. Accordingly, the study measured the influence of each of the OGD quality variables (data, system, and service quality) on citizens' trust in the data on open government platforms. Furthermore, this model helped estimate variables' relationships; the maximum likelihood was the best procedure to discover the strength and direction of the relationships connecting the proposed independent and dependent variables.

### 3.5. Construct Validity

In order to reach significant reliability and construct validity for all scales, confirmatory factor analysis was applied through the Analysis of Moment Structures (AMOS) program (Version 22). Through this model, all four constructs were permitted to freely co-vary, with each item acting as an indicator reflecting its latent construct. The maximum likelihood approach was employed to perform model estimation using the item correlation matrix as an input. The findings of this model, along with Cronbach's alphas and composite reliabilities, are presented in Table 3.

**Table 3.** Confirmatory factor analysis results.

| Item | Mean | Loading | Cronbach's Alpha | Composite Reliability |
|------|------|---------|------------------|------------------------|
| DQ_1 | 3.98 | 0.954 | 0.88 | 0.90 |
| DQ_2 | 3.78 | 0.861 | | |
| SYSQ_1 | 3.97 | 0.797 | | |
| SYSQ_2 | 3.92 | 0.752 | 0.91 | 0.89 |
| SYSQ_3 | 4.29 | 0.848 | | |
| SYSQ_4 | 4.23 | 0.891 | | |
| SERQ_1 | 3.69 | 0.815 | | |
| SERQ_2 | 4.21 | 0.782 | 0.85 | 0.88 |
| SERQ_3 | 3.98 | 0.795 | | |
| SERQ_4 | 3.26 | 0.839 | | |
| TR_1 | 3.72 | 0.851 | | |
| TR_2 | 3.98 | 0.893 | 0.87 | 0.91 |
| TR_3 | 3.18 | 0.905 | | |

The model fit and causal relationships between constructs in the structural model were evaluated using data from the validated measures. As depicted in Table 4, this process found good model fit since all model fit indices were within the recommended values, with CMIN/DF = 2.33 (CMIN = 1355, df = 580), RMSEA = 0.052, CIF = 0.930, NNFI = 0.949, and AGFI = 0.891 [9,34–37]. Due to the problems created by multicollinearity, these findings were checked, showing that the values of the variance inflation factor (VIF) were in the range of 1.35–1.82 for all constructs, which is considered acceptable.

**Table 4.** Measurement model fit indices.

| Fit Index | Result | Recommended Criteria | Source |
|-----------|--------|----------------------|--------|
| CMIN/DF ($\chi^2$/DF) | 1355/580 = 2.33 | $\leq 5$ | [32] |
| RMSEA | 0.052 | $\leq 0.08$ | [38] |
| CIF | 0.930 | $\geq 0.90$ | [37] |
| NNFI | 0.949 | $\geq 0.90$ | [39] |
| AGFI | 0.891 | $\geq 0.80$ | [39] |

The convergent validity of the measurement scales was verified according to the three criteria described by [40]. These criteria demand that (1) all indicator loadings must surpass the threshold value of 0.7, (2) construct reliabilities must be greater than 0.8, and (3) the average variance extracted (AVE) for each construct must be significant and greater than its variance, i.e., greater than 0.5. As shown in Table 3, all loadings in this model exceeded the threshold of 0.7, the composite reliabilities of all constructs were between 0.85 and 0.91, and AVE values were between 0.61 and 1.00. Therefore, the three criteria for convergent validity were all met. Additionally, discriminant validity was evaluated following Fornell and Larcker's recommendation that the square root of the average variance extracted from a certain construct must exceed the value of the correlation between this construct and others included in the same model [40]. The list of correlations between all constructs is given in Table 5, along with the square root of the AVE. Each diagonal value shown in bold

represents the square root of its construct's average explained variance. The values under each diagonal value represent the correlations between the related constructs, and each diagonal value was found to be greater than the correlations between related constructs. Therefore, the assessment of the discriminant validity was acceptable and verified for these constructs.

**Table 5.** Inter-item correlations.

|  | AVE | Data Quality | System Quality | Service Quality | Trust |
|---|---|---|---|---|---|
| Data Quality | 0.82 | **0.91** | | | |
| System Quality | 0.67 | 0.58 | **0.82** | | |
| Service Quality | 0.65 | 0.43 | 0.49 | **0.81** | |
| Trust | 0.78 | 0.56 | 0.43 | 0.41 | **0.88** |

Note: Square root of AVE shown in bold as the diagonal.

## 4. Results

The results of the structural model evaluation were based on investigating the anticipated capabilities of the proposed model and the relationships between its constructs, as indicated in Figure 2. This assessment involved issues of collinearity, the significance and relevance of path coefficients, and values of $R^2$ to specify the variance of a construct. In order to avoid collinearity issues, the VIF values were checked. Values over 5.00 would indicate problems associated with collinearity, meaning some indicators would have to be eliminated. However, the VIF values were all below 5.00 for all predictors. Moreover, values of $R^2$ for latent variables are between 0 and 1, with 0.25 being weak, 0.50 moderate, and 0.75 substantial [41]. The $R^2$ for data quality (DQ) was 0.449 and for system quality (SYSQ) was 0.469, which would be considered moderate. However, the $R^2$ for trust (TR) of 0.302 would be considered weak.

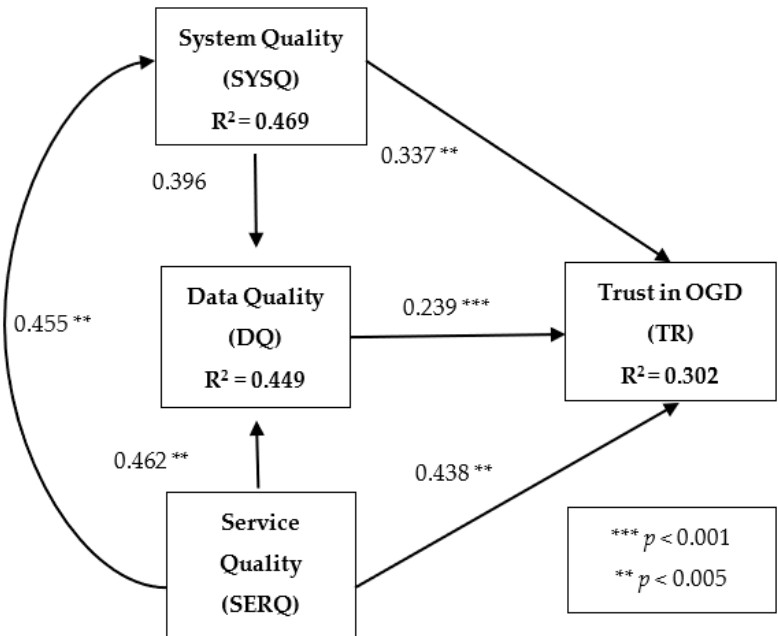

**Figure 2.** Structural model.

The significance and relevance of the path coefficients were evaluated. Path coefficients interpret how powerful an impact one variable has on another, with their weights making it possible to determine their relative significance. The significance level of a path coefficient is 5% when using a two-tailed *t*-test [41,42]. In evaluating their relevance, path coefficients should take values between −1 and +1. Whenever the value of the path

coefficient comes closer to +1, it demonstrates strong positive relationships, and when it is closer to −1, it indicates strong negative relationships [41]. The path coefficients of hypothesized relationships between variables for this research model and *t*-values for all hypotheses are presented in Table 6.

**Table 6.** Path coefficients.

| Hypothesis | Path | Estimate | *t* |
|:---:|:---:|:---:|:---:|
| 1 | DQ→TR | 0.239 *** | 4.55 |
| 2 | SYSQ→TR | 0.337 ** | 3.88 |
| 3 | SYSQ→DQ | 0.396 *** | 4.50 |
| 4 | SERQ→TR | 0.438 ** | 6.39 |
| 5 | SERQ→DQ | 0.462 ** | 4.78 |
| 6 | SERQ→SYSQ | 0.455 ** | 3.52 |

*** $p < 0.001$. ** $p < 0.005$.

The first hypothesis was supported, as perceived data quality had a significant impact on citizens' trust in OGD ($\beta = 0.239$, $p < 0.001$, $t = 4.55$). Likewise, perceived trust in OGD was significantly affected by system quality ($\beta = 0.337$, $p < 0.005$, $t = 3.88$) and service quality ($\beta = 0.438$ $p < 0.005$, $t = 6.39$), supporting the second and fourth hypotheses. Furthermore, system quality ($\beta = 0.396$, $p < 0.001$, $t = 4.50$) and service quality ($\beta = 0.462$, $p < 0.005$, $t = 4.78$) had a significant impact on perceived data quality, supporting the third and fifth hypotheses. Finally, service quality had a significant impact on system quality ($\beta = 0.455$, $p < 0.005$, $t = 3.52$), supporting the sixth hypothesis. Accordingly, all discussed relationships between constructs in the structural model were statistically significant, and service quality was found to have a significant influence on all other constructs represented in perceived data quality, system quality, and citizens' trust in OGD.

## 5. Discussion

### 5.1. Citizens' Willingness to Trust OGD

This study investigated whether citizens' trust in OGD would be affected by their perception of data, system, and service quality. The sample included only respondents with experience using OGD, and approximately 75% had used OGD for five years or more. Although studies have found repeated barriers to proper OGD use (e.g., [43–45]), many people have enjoyed using OGD and use it to create useful applications [46] or participate in open data innovation [47]. Trust is generally built through experience: in this study, the experience OGD users had developed through repeated use of OGD shaped their trust in it [3,4].

### 5.2. Information Systems Success Model Facilitate Citizens' Trust in OGD

The evaluation of loadings and reliability using confirmatory factor analysis indicated that DQ1, SYSQ4, and SERQ4 were the most significant attributes in perceived data, system, and service quality. DQ1 was significant because it investigated engagement with OGD and showed the importance that data are error-free. In the same manner, SYSQ4 evaluated the level of engagement with OGD systems and revealed that users found guidance from the system on how to interpret and download data. SERVQ4 was significant because it showed the importance of OGD services being user-centered. Some studies have addressed barriers to citizens' trust in OGD and their willingness to use it, such as errors in OGD [2,4]. Users of OGD systems should not face any challenges in downloading, interpreting, or providing feedback on data [7,48,49]. In addition, some practices from service providers were considered biased and seen as not effectively and efficiently paying attention to citizens' needs, causing user dissatisfaction [4,9,13].

The findings conform to prior research by emphasizing the role of data completeness in engagement with OGD (e.g., [9,47]). The OGD systems were found to have effective performance and functionality that facilitated engagement by providing data visualization



and enabling feedback and quality rating. In this study, OGD users confirmed receiving sufficient responses from OGD providers in a timely manner. Unlike most of the studies on OGD and user trust that found a strong link between perceived data quality and trust, the findings of this study revealed that perceived service quality was the most significant factor driving citizens' trust in OGD. This variation might be attributed to users who have different nationalities, levels of education, and experiences with OGD systems and services. Most respondents were highly educated with majors in science, computer science, or education, and about 75% had used these systems for over five years. This is expected to give users more ability to differentiate between high- and low-quality data, provide sound feedback, detect errors, and request actions to correct biased or incomplete data.

Regarding the importance of OGD service quality and its impact on citizens' trust, a study in Austria revealed that allowing citizens to share their expertise, opinions, and innovative ideas was a good way of building trust between government and citizens [3]. Moreover, it is important for OGD providers to respond quickly to user requests to enhance the data provided. These providers should be ready to discuss data and follow up with citizens to correct misunderstandings or incorrect data [50]. When OGD providers show attention to users' requests and provide high-quality support, users become more satisfied with the service than with the quality of the data themselves [51]. The present study confirmed that, unlike OGD users' perceptions of system quality, higher perceived service quality was significantly linked to higher perceived data and system quality.

This study is one of the few to investigate the impact of perceived data, system, and service quality on citizens' perceived trust in OGD. All hypotheses were supported, indicating citizens' perceived trust in OGD, which was significantly affected by the perceived quality of the data, system, and services. This trust allows users, especially those with the knowledge and skills to interact with open government systems, to benefit from OGD.

## 6. Conclusions

This study sets the basis for how effectively governments can encourage citizens' trust in OGD. The results suggested that when citizens perceived that there to be higher data, system, and service quality, they were more likely to trust OGD. The information systems success model was employed, a model not commonly used with open government and OGD, by integrating the three quality attributes of OGD (i.e., data, system, and service quality).

The study analyzed 358 complete responses of citizens from different nationalities, representing 63.58% of all responses. About 75% of respondents had used OGD for over five years, suggesting extensive experience with open government systems, data, and services. After all complete responses were analyzed using a structural model and the relationships between constructs were examined, all hypotheses were supported. This indicated that citizens' trust in OGD could be anticipated through their perception of data, system, and service quality. Moreover, service quality had a positive impact on the perceived quality of data and the system, and perceived system quality had an influence on perceived data quality. The strong influence of service quality on perceived data quality and trust in OGD could serve as an incentive for OGD providers to improve their services by offering tools for feedback and data visualization. The more effective and accessible support was, the more important the system was for new and experienced users alike. This study maintains that the perceived quality of data, systems, and services provided by the open government are key components of citizens' perceived trust in OGD.

This paper provides evidence of the significance of citizens' trust in OGD and the need for increasing this trust. However, trust in OGD is not built overnight, and it is the responsibility of open governments to offer high-quality data, systems, and services. In addition, open governments should empower citizens by enabling them to contribute. This study could serve as an empirical reference for researchers assessing the impact of data, system, and service quality on citizens' perceived trust in OGD.

The focus of this study was on how users perceived the quality of OGD, systems, and services. However, the type of data available, the systems used, and the services provided were not investigated. Accordingly, further research could address these issues and other variables influencing citizens' trust in OGD, such as disclosure of information and the relationship between government and citizens. Furthermore, the level of citizens' trust in OGD along with the information systems success model could be examined in the context of local government.

**Author Contributions:** Conceptualization, A.A. (Abdullah Almuqrin), I.M., M.A. and A.A. (Abdulaziz Alomran); methodology, A.A. (Abdullah Almuqrin), J.G. and I.M.; formal analysis, A.A. (Abdullah Almuqrin), I.M. and M.A.; writing—original draft preparation, A.A. (Abdullah Almuqrin), J.G., M.A. and A.A. (Abdulaziz Alomran); writing—review and editing, I.M., A.A. (Abdullah Almuqrin), J.G. and M.A. All authors have read and agreed to the published version of the manuscript.

**Funding:** This research was funded by the Researchers Supporting Project number (RSP2022R453), King Saud University, Riyadh, Saudi Arabia.

**Institutional Review Board Statement:** The study was conducted in accordance with the Declaration of Helsinki, and approved by the Institutional Review Board (Human and Social Researches) of King Saud University (Ref No: KSU-HE-18-242).

**Informed Consent Statement:** Informed consent was obtained from all participants involved in this study. Therefore, an online agreement that all participants agree to by checking a box that says "I agree" before starting the questionnaire.

**Data Availability Statement:** Not applicable.

**Acknowledgments:** This research was funded by the Researchers Supporting Project number (RSP2022R453), King Saud University, Riyadh, Saudi Arabia.

**Conflicts of Interest:** The authors declare no conflict of interest.

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
