# Peer review of "Factors Influencing Public Trust in Open Government Data"

_sustainability, doi:10.3390/su14159765_

Round 1

Reviewer 1 Report

The article examines the impact of data perception, system, and service quality on citizens’ trust in Open Government Data (OGD). In the opinion of the reviewer, the manuscript is a theoretical and empirical study that addresses current challenges in the area of OGD. Detailed article assessment:

1)    The theoretical background and the selection of references are consistent with the topic and empirical research.

2)    The study is well-designed. In the methodological section, the research procedure was described in detail, and the hypotheses were correctly formulated.

3)    The results of the research are described in detail. Statistical analysis was used to present the research results. The conclusions are justified and supported by research results.

In the reviewer’s opinion, the article is a valuable and interesting study. Research results can be helpful in designing ODG solutions.

Author Response

Responses to Reviewer 1

Minor English changes required

This point has been addressed in the manuscript

Reviewer 2 Report

Using "DeLone and McLean’s information systems success model" framework is an interesting frame of reference.

However, while analyzing the data the author make use of "Anderson and Gerbing’s two-step approach".

I would like authors to comment on the analytical incompatibilities that may arise later in the outcomes that are based on a data derived from Likert Scale Survey. Some readers may find it difficult to digest convergent validity of the measurement scales verification based on 3 criteria authors have used in their analysis.

Overall paper is good, however, aligning analytical framework and making sure there are no theoretical anomalies will enhance the value of the paper.

Author Response

Responses to Reviewer 2

Moderate English changes required

This point has been addressed

Using "DeLone and McLean’s information systems success model" framework is an interesting frame of reference. However, while analyzing the data the author make use of "Anderson and Gerbing’s two-step approach".

The DeLone and McLean’s information system success model was used to create the constructs of this study; however, the Anderson and Gerbing’s two-step approach was used to evaluate the validity of the constructs and examine the relationships between them. The approach was used by many studies [1, 2, 3, 4]

I would like authors to comment on the analytical incompatibilities that may arise later in the outcomes that are based on a data derived from Likert Scale Survey.

This point has been addressed in the manuscript.

Some readers may find it difficult to digest convergent validity of the measurement scales verification based on 3 criteria authors have used in their analysis.

The 3-criteria of the convergent validity are the best way to inspect convergent validity. However, your opinion might be true for non-specialized readers. Therefore, this was considered in the paper by using academic language that is easy to read and understand.

Overall paper is good, however, aligning analytical framework and making sure there are no theoretical anomalies will enhance the value of the paper.

The analytical framework was based on the main factors proposed by Delone and McLean in their information system success model. These factors aimed to providing net benefits which is represented by trust in the open government data in this study. The analysis of the framework components proved to be effective in enhancing the trust of IS users in OGD.

References

[1] Jöreskog, K., & Sörbom, D. (1993). LISREL 8: Structural equation modelling with the SIMPLIS command language. Scientific Software International.

[2] Kim, S., Sturman, E., Kim, E.S. (2015). Structural Equation Modeling: Principles, Processes, and Practices. In: Strang, K.D. (eds) The Palgrave Handbook of Research Design in Business and Management. Palgrave Macmillan, New York. https://doi.org/10.1057/9781137484956_11

[3] Arasavilli, A., & Babu, M. K. (2021). Evaluating factors that impact student decisions on higher education abroad: A structural equation modelling approach. Webology, 18(2), 1474-1485.

[4] Kautish, P., Khare, A., & Sharma, R. (2021). Influence of values, brand consciousness and behavioral intentions in predicting luxury fashion consumption. The Journal of Product & Brand Management, 30(4), 513-531. https://doi.org/10.1108/JPBM-08-2019-2535

Reviewer 3 Report

The paper aims to increase understanding of the relationships between perceived quality of data, systems, and services on citizens’ trust in  Open Government Data (OGD). The authors use structural equation modeling to conduct path analysis of the constructs. The authors find several interesting relationships, such as OGD service quality affects data and system quality.

The writing and the presentation of the paper are clear, the structure is adequate. However, I would like to point out some suggestions to improve the fluidity and compression of the text:

Point 1) The subsection 1.1 should be in a new section called: "Theoretical Framework" or "Background". This change would make the text flow more understandable. 

Point 2) Please, I suggest to highlight the innovative contribution of the work in the introduction and a paragraph (before the "Theoretical Framework") defining and describing briefly the independent variables analysed in the work: data, system, and service quality.

Point 3) It is not clear why the authors are using the DeLone and McLean's model in the theoretical framework.

Point 4) The statement in line 139 "Service quality, especially in marketing, has five major dimensions: assurance, empathy, responsiveness, reliability, and tangibles" is not fully true. This is the Parasuraman model (SERVQUAL) for service quality. There are other models that encodes other dimensions, especially for digital public services. My suggestions is to cite other models/works in this part.

Point 5) Please, provide more information about the instrument (questionnaire) developed in section 2.1. Which languages the questionnaire was developed? How long on average it would take a user to answer the questionnaire? Is it feasible to put the full questionnaire in the paper?

Point 6) The text of the Discussion section should be divided into two subsections, each one related to the two questions presented in the Introduction section: "Why are citizens willing.." and "How can the information systems success..."

Point 7) The word "evidence" in the Conclusion is very strong, as the work was developed over a convenience sampling. This fact may not be true in other contexts.

Author Response

Responses to Reviewer 3

Minor English changes required

This point has been addressed in the manuscript

The subsection 1.1 should be in a new section called: "Theoretical Framework" or "Background". This change would make the text flow more understandable. 

This point has been addressed in the manuscript

Please, I suggest to highlight the innovative contribution of the work in the introduction and a paragraph (before the "Theoretical Framework") defining and describing briefly the independent variables analysed in the work: data, system, and service quality.

This point has been addressed in the manuscript

It is not clear why the authors are using the DeLone and McLean's model in the theoretical framework.

The DeLone & McLean Information System Success Model was used because it addresses the influence of data, system, and service quality of an IS on net benefits delivered to system users. I found that these constructs are the most appropriate for gaining trust in OGD and therefore I used it in the theoretical framework.

The statement in line 139 "Service quality, especially in marketing, has five major dimensions: assurance, empathy, responsiveness, reliability, and tangibles" is not fully true. This is the Parasuraman model (SERVQUAL) for service quality. There are other models that encodes other dimensions, especially for digital public services. My suggestions is to cite other models/works in this part.

This point has been addressed in the manuscript

Please, provide more information about the instrument (questionnaire) developed in section 2.1. Which languages the questionnaire was developed? How long on average it would take a user to answer the questionnaire? Is it feasible to put the full questionnaire in the paper?

This point has addressed in the manuscript

However, the full questionnaire will be used in another current research and will be available with it.

The text of the Discussion section should be divided into two subsections, each one related to the two questions presented in the Introduction section: "Why are citizens willing.." and "How can the information systems success..."

This point has been addressed in the manuscript

The word "evidence" in the Conclusion is very strong, as the work was developed over a convenience sampling. This fact may not be true in other contexts.

This point has been addressed in the manuscript

Round 2

Reviewer 3 Report

All my suggestions have been taken into account by the authors in this new version of the article.